

# Nudging allows direct evaluation of coupled climate models with in-situ observations: A case study from the MOSAiC expedition

Felix Pithan[1], Marylou Athanase[1], Sandro Dahlke[1], Antonio Sánchez-Benítez[1], Matthew D. Shupe[2,3], Anne Sledd[2,3], Jan Streffing[1,4], Gunilla Svensson[5], and Thomas Jung[1,6]

[1]Alfred Wegener Institute, Helmholtz Centre for Polar and Marine Research (AWI), Bremerhaven/Potsdam, Germany
[2]Cooperative Institute for Research in Environmental Sciences, University of Colorado Boulder, Boulder, Colorado
[3]National Oceanic and Atmospheric Administration Physical Science Laboratoriy, Boulder, Colorado
[4]Jacobs University Bremen, Bremen, Germany
[5]Department of Meteorology and Bolin Centre for Climate Research, Stockholm University, Stockholm, Sweden
[6]Institute of Environmental Physics, University of Bremen, Bremen, Germany

**Correspondence:** Felix Pithan (felix.pithan@awi.de)

**Abstract.** Comparing the output of general circulation models to observations is essential for assessing and improving the quality of models. While numerical weather prediction models are routinely assessed against a large array of observations, comparing climate models and observations usually requires long time series to build robust statistics. Here, we show that by nudging the large-scale atmospheric circulation in coupled climate models, model output can be compared to local observa-
tions for individual days. We illustrate this for three climate models during a period in April 2020 when a warm air intrusion reached the MOSAiC expedition in the central Arctic. Radiosondes, cloud remote sensing and surface flux observations from the MOSAiC expedition serve as reference observations. The climate models AWI-CM1/ECHAM and AWI-CM3/IFS miss the diurnal cycle of surface temperature in spring, likely because both models assume the snow pack on ice to have a uniform temperature. CAM6, a model that uses three layers to represent snow temperature, represents the diurnal cycle more realisti-
cally. During a cold and dry period with pervasive thin mixed-phase clouds, AWI-CM1/ECHAM only produces partial cloud cover and overestimates downwelling shortwave radiation at the surface. AWI-CM3/IFS produces a closed cloud cover but misses cloud liquid water. Our results show that nudging the large-scale circulation to the observed state allows a meaningful comparison of climate model output even to short-term observational campaigns. We suggest that nudging can simplify and accelerate the pathway from observations to climate model improvements and substantially extends the range of observations
suitable for model evaluation.

## 1 Introduction

As any model, a model of the Earth's atmosphere is not an exact copy of what it represents (Box, 1979). To make the best possible use of a model for research as well as for scenarios or forecast applications, it is important to understand the degree and the limitations to which a weather or climate model truthfully represents the physics that govern the real system. Comparing
model output with real-world observations is crucial to obtain such understanding (Eyring et al., 2019). For numerical weather prediction models, this happens routinely in forecast verification, as forecasts of the atmospheric state are compared to the state



that actually occurred (Casati et al., 2008). The reduction of forecast errors can be tracked from model version to model version or over decades, and the forecasting capability has continually increased by about one day per decade (Bauer et al., 2015).

Comparing the output of a coupled atmosphere-ocean climate model to observations is less straightforward. The purpose of
a climate model is to reproduce the long-term average state and the variations of the Earth's climate system given an external forcing, such as the orbital configuration, greenhouse gas and ice sheet extent of the last glacial maximum or of today's climate. Even on decadal time scales, any given local, regional or global observation is subject to substantial internal variability, such that even perfect models would not exactly reproduce the observable (Notz, 2015). Large spatial and temporal scales are therefore required for model-data comparisons, which limits the amount of independent data points that can be used for model
evaluation. These datasets are often highly aggregated, rendering it difficult to use a mean state comparison between model and observations to infer something about the representation of a specific process in the climate model.

Process-based diagnostics enable a broader comparison of climate model data with short and high-frequency observational records and can help to reveal at least qualitative, categorical errors of climate models (Eyring et al., 2005; Ahn et al., 2017). They can also be useful to reveal biases that only occur in a specific state of the atmosphere.

A number of climate model setups have been developed to constrain the dynamics, i.e. the atmospheric circulation mostly in atmosphere-only models in order to directly compare the model physics, including thermodynamic processes, to observations. Single-column setups prescribe the horizontal advective tendencies and vertical motions at a given point or follow a column of air in Lagrangian simulations (Randall et al., 1996; Bretherton et al., 1999). The Transpose-AMIP (Atmospheric Model Intercomparison Project) approach effectively runs climate models in a weather forecast mode, initialising the atmosphere to
its observed state and studying the short-time evolution of the atmospheric state in models. Despite the risk of an initial shock when the model is started in a state that it might never generate by itself, this method has proven useful to diagnose the genesis of cloud biases in the Southern Ocean, for example (Williams et al., 2013).

Climate models can be nudged to the observed atmospheric circulation by relaxing the model state to reanalysis data at each timestep (Coindreau et al., 2007). Nudging can be restricted to certain regions, altitudes and variables. Analysing nudging
increments can allow to pinpoint where models tend to deviate from the processes occurring in the real atmosphere. This method has been applied to evaluate processes related to atmospheric dynamics and the momentum budget by van Niekerk et al. (2016). Wehrli et al. (2018) used nudging to demonstrate that an overestimation of hot, dry mid-latitude summers in CESM (Community Earth System Model) is largely caused by thermodynamic processes rather than a biased large-scale circulation.

Nudged climate models including coupled atmosphere-ocean models have recently also been used to study specific events such as heat waves across different climate states (van Garderen et al., 2020; Wehrli et al., 2020; Sánchez Benítez et al., 2022). In these studies, a given event is recreated by nudging the (large-scale) atmospheric circulation to its observed state, and the climate state can be altered by initialising the model in present, pre-industrial or possible future climates (Shepherd et al., 2018). Comparison of the present-day runs with observations have shown a close match on a day-to-day basis.

Here, we explore the possibility of using nudged runs of coupled atmosphere-sea ice-ocean models for evaluating model physics. In contrast to the Transpose-AMIP approach, a nudged coupled model is spun up over several months to one year and





then can be run for several years, such that any initial shock would not affect the model-data comparison. Nudging ensures that the model follows the observed trajectory of the atmospheric state over time, whereas transpose-AMIP setups strongly deviate from observations within days after their initialisation.

While the large-scale atmospheric circulation is constrained by the nudging, the thermodynamics of the climate system can entirely be left to the model itself, such that clouds, temperatures including ocean temperatures, sea ice and the water budget are fully computed by the model with no other constraints than the imposed large-scale winds in the free troposphere.

The approach is limited to observed phenomena that are strongly constrained by the large-scale vorticity and divergence. This includes many important meteorological phenomena in the extratropics from mid-latitude heat waves and cyclones to

intrusions of warm, moist air and cold-air outbreaks in the Arctic. Weather events in the Tropics or events in mid-latitudes that are driven by localised convection are probably more difficult to capture using this approach.

We use April 2020 observations from the MOSAiC (Multidisciplinary drifting Observatory for the Study of Arctic Climate) expedition as a case study (Shupe et al., 2020). During MOSAiC, the German research icebreaker Polarstern (Knust, 2017) drifted across the central Arctic Ocean from October 2019 to September 2020.

The beginning of April is characterized by cold, dry conditions in the central Arctic, whereas several warm, moist intrusions reach the observational site in mid April. Such intrusions are driven by the large-scale circulation (Woods et al., 2013; Pithan et al., 2018), which is constrained in our model setups. We can thus use in-situ observations at the MOSAiC site to evaluate how models handle the thermodynamic transformation of the initially warm and moist air mass that cools and loses moisture through precipitation over Arctic sea ice. This transformation depends on mixed-phase microphysics and surface interactions

that are challenging to represent in large-scale models and contributed to important Arctic climate biases in CMIP5 climate models (Pithan et al., 2014, 2016).

In the following, we briefly present the evaluated models, their representations of key physical processes and our nudging method as well as the observational datasets used. We then evaluate model output against observations for April 2020.

## 2  Models and data

### 2.1  Models

We use the Alfred Wegener Institute's coupled climate models AWI-CM1 (Sidorenko et al., 2015; Rackow et al., 2018) and AWI-CM3 (Streffing et al., 2022). In AWI-CM1, the FESOM 1.4 sea ice-ocean model (Wang et al., 2014) is coupled to the atmospheric model ECHAM6.3 (Stevens et al., 2013), while in AWI-CM3 FESOM2 [Cite: https://doi.org/10.5194/gmd-12-4875-2019 and https://doi.org/10.5194/gmd-15-335-2022] is coupled to OpenIFS 43r3 (ECMWF, 2017). Full documentations

of the models are available under the above references. We present aspects of both models that are especially relevant for our study period below, namely cloud processes, boundary-layer turbulence and surface coupling. For some analyses, we include data from an atmosphere-only run (i.e. prescribed sea-surface temperatures and sea-ice extent) with CAM6 (Danabasoglu et al., 2020).



The MOSAiC drift during April 2020 and nearby model grid points are shown in Fig. 1. The gaussian grid of AWI-
CM1/ECHAM results in a high zonal resolution close to the pole, whereas the reduced gaussian grid of AWI-CM3/IFS and the
cubed-sphere grid of CAM6 have substantially less grid points in the area. The ratio between the zonal and meridional spacing
of grid points is closer to unity in the latter models. For technical reasons, AWI-CM3/IFS data was re-gridded to a regular
1x1-degree grid before being analysed. Model time series are taken from the grid point closest to the MOSAiC observatory in
mid-April, as choosing the closest grid point for each time step did not lead to considerably different results.

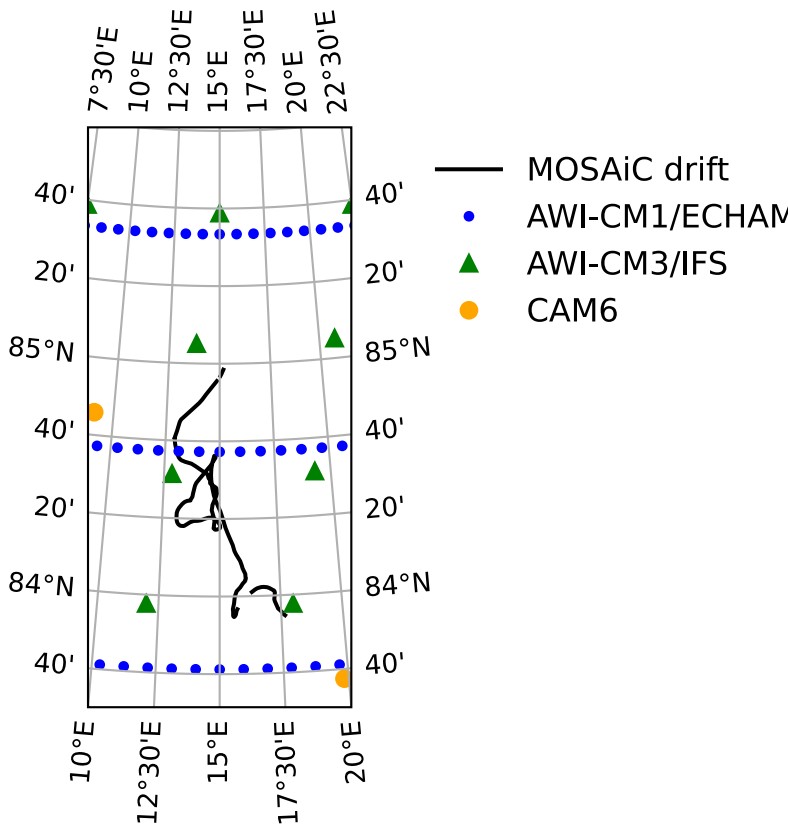

**Figure 1.** Drift of the MOSAiC central observatory during April 2020 and model grid points covering the area. The observatory is drifting
southward.

### 2.1.1 Cloud schemes


The presence of clouds in ECHAM is diagnosed following Sundqvist et al. (1989). Cloud fraction is parameterized as

$$f_{cloud} = 1 - \sqrt{\frac{rh_{sat} - rh}{rh_{sat} - rh_{crit}}}, \tag{1}$$



where $rh$ is the relative humidity in a model gridbox, $rh_{sat}$ the saturation relative humidity and $rh_{crit}$ the critical relative humidity, which is the threshold for cloud formation. $rh_{sat}$ is generally set to one and $rh_{crit} = 0.7 + 0.2 exp\left(1 - \left(\dfrac{p_{sfc}}{p}\right)^4\right)$,

where $p$ is pressure and $p_{sfc}$ is surface pressure. If a temperature inversion exists below the 700 hPa height, these parameters are set to $rh_{crit} = 0.7$ and $rh_{sat} = 0.9$.

Cloud water and cloud ice are treated as prognostic variables following Lohmann and Roeckner (1996).

In the IFS, cloud fraction is a prognostic variable that is changed by horizontal transport, detrainment from convection, large-scale condensation and evaporation (ECMWF, 2017). Large-scale condensation increases the cloud fraction whenever

the saturation specific humidity decreases, i.e. when an air parcel is cooling. Evaporation decreases the cloud fraction 1) when the saturation specific humidity increases and the cloud water content approaches zero and 2) by mixing with environmental air from the cloud free part of the grid box.

### 2.1.2 Boundary-layer turbulence and surface coupling

In ECHAM6, turbulent fluxes between the atmosphere and surface and within the atmosphere are computed using prognostic

turbulent kinetic energy to compute the turbulent diffusivities for heat and momentum (Brinkop and Roeckner, 1995). In the IFS, only the mean winds are prognostic variables and turbulence is diagnosed at each time step. Above the surface layer, the diffusivity approach is combined with a mass-flux scheme to represent the effect of large turbulent eddies in convective boundary layers (ECMWF, 2017). In CAM6, turbulent fluxes and shallow convection are computed in the CLUBB scheme. CLUBB prognostically computes sub-grid variances and co-variances to determine the turbulent fluxes (Larson, 2022; Guo

et al., 2021).

While the IFS uses a skin temperature that can be different from the surface temperature and a skin layer conductivity dependent on surface type and conditions, ECHAM6 has no separate skin layer and directly uses the temperature of the uppermost surface layer to compute fluxes. In both models, the temperature of the uppermost layer represents the uppermost 10 cm of sea ice plus the entire snow layer. This surface temperature is updated every time step (200s) inside the atmospheric

model in AWI-CM1/ECHAM, but only at every coupling step (2h) in AWI-CM3/IFS, where the temperature update occurs within the sea-ice model.

### 2.2 Nudging

In the AWI models, vorticity and divergence in the free troposphere (700 to 200 hPa) are nudged to ERA5 reanalysis (Hersbach et al., 2020) using a spectral truncation of T20. AWI-CM3 is nudged with a 1-hour relaxation time scale, which results in

similar mean values and a smaller spread of atmospheric temperature profiles compared to a 24-hour relaxation timescale in the present case study (not shown). An even stronger nudging setup without truncation, i.e. nudging all wavenumbers of vorticity and divergence in AWI-CM3, did not further reduce the spread between ensemble members but lead to a stronger cold bias development in the Arctic on interannual time scales. We use AWI-CM1 runs with a relaxation time-scale of 24 hours that were originally produced for a study on European heatwaves, where the longer relaxation timescale allowed a good match





with reanalysis data (Sánchez Benítez et al., 2022). We do not expect the different relaxation timescales to impact our results beyond the larger ensemble spread in AWI-CM1. Five ensemble members for each model are initialised on 1st January 2017 from different atmosphere and ocean states based on CMIP6 ssp370 scenario forcing (O'Neill et al., 2016).

In the uncoupled, i.e. atmosphere-only, CAM6 run used here, free-tropospheric (above 690 hPa) temperature and horizontal wind components are nudged using hourly updated ERA5 fields. Note that the wind field is not truncated in this non-spectral
model, such that the full field including smaller scales is used for nudging. Daily values of sea-ice concentration and SST are interpolated from monthly HadISSTdata. We focus on evaluating the near-surface variables from this run, where we expect physical errors from boundary-layer and cloud processes to dominate with minimal impacts of the different setup.

## 2.3   Observations

We use observational data from the MOSAiC Central Observatory (Shupe et al., 2022). On the sea-ice adjacent to Polarstern,
measurements of near-surface temperature, wind speed, snow physical depth, and sensible heat flux were made from a 10-m meteorological tower (Cox et al., 2021). Near the tower, a suite of up- and down-looking broadband radiometers measured the incident and reflected solar radiation (Riihimaki, 2019) and were used to derive the surface skin temperature. All of these on-ice measurements were representative of a relatively small domain directly around the measurements in questions, often representing domains on the order of 1 up to  100 meters. Onboard Polarstern, measurements from a ceilometer (Morris
et al., 2021) provided information about cloud occurrence. Cloud microphysical properties were derived from multiple ship-based sensors including radar, lidar, microwave radiometer, ceilometer, and radiosondes using the ShupeTurner cloud retrieval algorithm (Shupe et al., 2015; Shupe, 2022). Radiosondes were launched 4-7 times per day during the period of interest from the back deck of Polarstern, providing profiles of atmospheric state variables (Maturilli et al., 2021). While all of these ship-based measurements were made in a vertical or slant-path above Polarstern, we assume they are representative of the domain
directly adjacent to Polarstern as well, including the other on-ice measurements.

Observed snow temperatures were taken from 10 Snow and Ice Mass Balance Arrays (SIMBA), installed during the fall and spring of MOSAiC at the Central Observatory as part of the Distributed Network (Lei et al., 2022). SIMBAs are a 5 m long thermistor chain with sensors at 2 cm intervals installed vertically through the upper ocean, sea ice and snow, and lower atmosphere. Conductive fluxes are derived by assuming the temperature changes in time at a particular level in the
column are equal to the divergence of vertical conduction and extinction of penetrating solar radiation, as in Lipscomb (1998). Profiles of thermal conductivity are solved for using temperature profiles from SIMBAs starting from 100 cm below the sea ice-snow interface through the snow top. Solar radiation is assumed to decay exponentially in sea ice and snow, a constant bulk extinction coefficient of 1.5 m$^{-1}$ is assumed for sea ice. A 7-day running mean derived from light chain buoy 2020R11 following Katlein et al. (2021) is used for the bulk extinction coefficient of snow. Density is assumed to be related to thermal
conductivity following Calonne et al. (2019). Heat capacity is assumed to depend on temperature (Paterson and Bryce, 1994). At the lower boundary, the thermal conductivity of sea ice is assumed to be 2 Wm$^{-1}$K$^{-1}$. For time steps with conductive flux convergence, i.e. when the vertical temperature gradient changes sign, derived thermal conductivities are unrealistically low.



For these time steps, thermal conductivity is interpolated from surrounding timesteps at each level. Conductive fluxes, C, are then re-calculated as

$$C = -k\frac{dT}{dz},$$
(2)

where k is thermal conductivity and $\frac{dT}{dz}$ is the vertical temperature gradient. Finally, the conductive heat flux is averaged across individual SIMBAs. Note that all fluxes are defined as positive towards the surface, i.e. downwards in the atmosphere and upwards through sea-ice and snow in agreement with climate modelling conventions.

Our study is focused on April 2020, which corresponds to a targeted observation period initiated by the Year of Polar
Prediction (YOPP) Process Task Team (Werner et al., 2020). During this period, several pulses of warm, moist air from the open ocean at lower latitudes reached the MOSAiC site, causing temperatures to rise close to the melting point.

## 3 Results

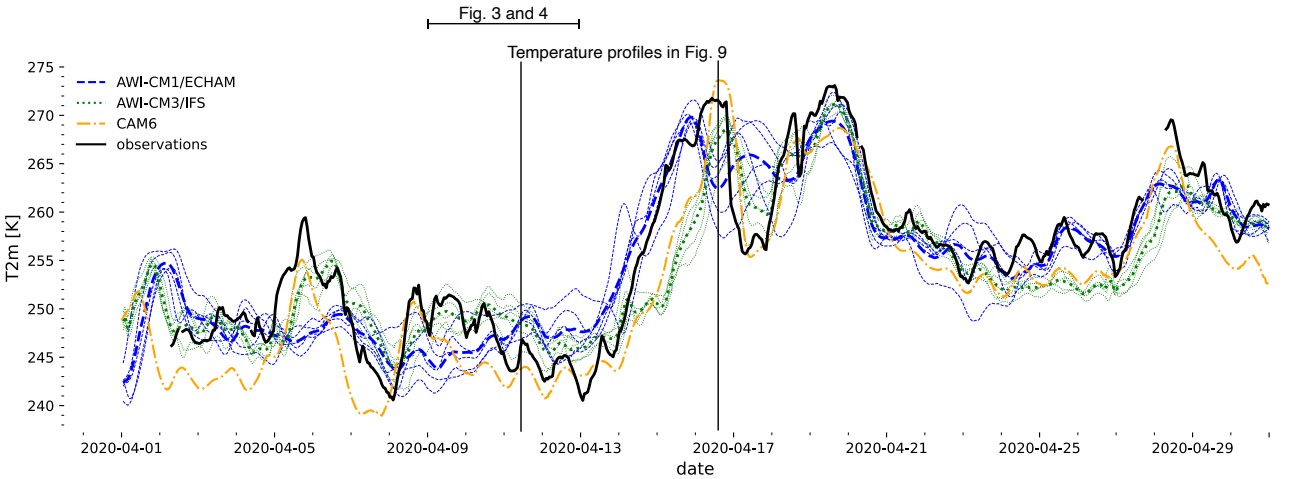

**Figure 2.** Observed and modelled hourly 2m temperatures during April 2020 at the MOSAiC site. Thin dashed and dotted lines show individual ensemble members, the thicker lines show ensemble mean for each AWI-CM model version. The period 9 to 12 April that we investigate in more detail and the timing of two soundings we analyse are indicated for reference.

During the first half of April, near-surface air temperature at the MOSAiC site was about 25 K below the freezing point, with a brief warming of about 10 K around 6 April (Fig. 2). The analysed models generally reproduce the observed temperature, but
AWI-CM1 misses the warming on 6 April. Both AWI models miss or substantially underrepresent the cooling trend and diurnal cycle visible in observations from 9 to 12 April, a period that is discussed in more detail below. CAM6 has a more realistic representation of the diurnal cycle and cooling trend over these days. Ensemble spread for AWI-CM1/ECHAM is larger than





for AWI-CM3/IFS due to the longer nudging timescale (24h vs. 1h). The ensemble spread for AWI-CM3/IFS, i.e. the strong nudging configuration, is substantially smaller than the differences between models or between models and observations at
most times. Such differences are thus robust to the remaining variability of the nudged model.

From 13 to 16 April, warmer and moister air masses from lower latitudes arrive at the MOSAiC site. Observed near-surface temperatures rise by about 25 K and thus reach close to the freezing point. The overall warming is reproduced by the models, with some delay in AWI-CM3 and CAM6. Observed temperature drops rapidly by about 15 K on 17 April as the MOSAiC region comes under the influence of colder air masses. As a second pulse of warm air arrives on 19 April, the temperature rises
to the freezing point again. After the moist intrusions, from 21 April onwards, the temperature stabilizes around 15 K below freezing, about 10 K warmer than before the intrusions. This stabilisation at a higher level is captured by all models.

AWI-CM3/IFS has the onset of the first pulse of the intrusion and some other (but not all) notable features of the temperature evolution delayed by about one day. We attribute this to the coarser horizontal resolution in the region compared to AWI-CM1/ECHAM (see Fig. 1).
We first analyse the surface (skin) temperature and components of the surface energy budget in the days prior to the moist intrusion and then temperature and cloud profiles before and during the intrusion.

## 3.1 Diurnal cycle of surface temperature and surface energy budget

The skin temperature evolution in the period 9 to 12 April is characterized by a pronounced diurnal cycle with a magnitude of about 5 K and a general cooling trend with a similar magnitude (see Fig. 3a). Neither the cooling nor the diurnal cycle
are realistically represented in the skin temperature of AWI-CM1/ECHAM and AWI-CM3/IFS, despite a diurnal cycle with realistic magnitudes in the total surface energy budget (see Fig. 3b). In contrast, a diurnal cycle is apparent in CAM6 surface temperature (orange dash-dotted line in Fig. 3a).

The unresponsiveness of modelled skin temperature to the diurnal cycle in the surface energy budget is probably due to the simplistic treatment of the snow pack on sea ice in both versions of AWI-CM. Snow temperature is assumed to be uniform
throughout the snow pack, which leads to a substantial thermal inertia. With a modelled snow thickness of about 0.1m water equivalent, a surface flux imbalance on the order of $100 \, \mathrm{Wm}^{-2}$ during one hour would be required to raise the temperature of the snowpack by 1 K. While the observed snow depth at the MOSAiC site in April was of similar magnitude (Wagner et al., 2021), flux imbalances that are an order of magnitude smaller are sufficient to raise the surface temperature by several degrees during the day. This points to a much thinner layer of snow being directly thermally coupled to the atmosphere, with the low
conductivity of snow limiting the vertical distribution of heat within the snowpack. Fig. 4 indeed shows that the diurnal cycle of temperature is strongest in the uppermost 2 cm of the snowpack (solid line) and substantially dampened below (dashed and dotted lines).

The diurnal cycle of temperature can have important implications for the surface albedo, as snow that melts during the day and refreezes at night has a different albedo than a snowpack with a temperature that consistently remains below freezing.
While temperatures remain well below freezing in the period discussed here, the models' inability to reproduce the observed diurnal cycle is thus a cause for concern, and fundamentally supports the idea to introduce more sophisticated representations of







**Figure 3.** Observed and modelled skin temperature, surface energy budget, shortwave fluxes, albedo and cloud cover for 9 to 12 April 2020, the cold period prior to the moist intrusion.





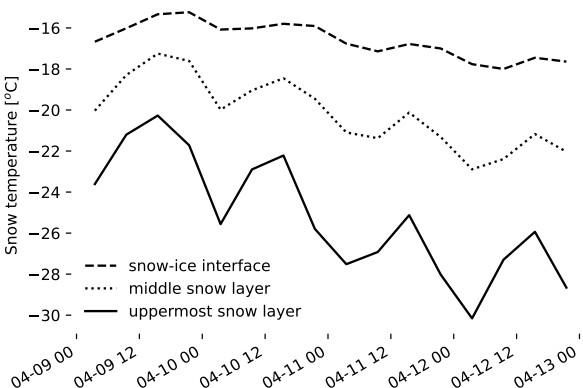

**Figure 4.** Temperature evolution within the snowpack at the snow-ice interface, 10 cm depth and just below the snow surface for the cold period 9 to 12 April 2020. Temperature is averaged from a subset of 4 SIMBAs with snow depths of 20-25 cm .

sea-ice and snow thermodynamics in climate models (Zampieri et al., 2021). However, thermodynamically more sophisticated models have not been shown to better reproduce observed sea ice trends or variability in the past (Blockley et al., 2020), suggesting that this is (or at least was) not a first-order problem in the CMIP5 generation of climate models.

Surface downwelling shortwave radiation tends to be overestimated by AWI-CM1 and matches observations in AWI-CM3 and CAM (Fig. 3c). However, absorbed shortwave radiation is underestimated by AWI-CM3 and slightly overestimated by AWI-CM1 (see Fig. 3d) due to the higher surface albedo in AWI-CM3 than in AWI-CM1 or the observations (Fig. 3e). In CAM6, the albedo closely matches the observed value during the first days and is somewhat lower thereafter. Note that we compare a 24-hour rolling average of the albedo, as the diurnal cycle in the albedo computed from observations is probably

overestimated (not shown). We attribute this to a slight shift between the timings of maximum upwelling and downwelling radiation, which might be caused by a sloped snow surface under the radiation sensors. Using a time-average not only smoothes this out, but also reduces the physically realistic time-dependence of the albedo that may be due to the sun angle or cloud state. Using the time-averaged albedo to recompute the net shortwave radiation at the surface (gray line in Fig. 3d) leads to a substantial shift in the diurnal cycle on 11 and 12 April, but does not affect our general conclusions: AWI-CM1 overestimates

and AWI-CM3 slightly underestimates absorbed shortwave radiation, whereas CAM6 closely matches observations.

### 3.2   Cloud cover, ice and liquid water

AWI-CM3 produces a virtually closed cloud cover throughout the discussed period, whereas AWI-CM1 produces more variable and lower cloud cover fractions (see Fig. 3f). Ceilometer data from MOSAiC indicates that clouds were detected in 58 % of all measurements from 9 to 12 April 2020 (Morris et al., 2021), but the cloud radar detects cloud condensate almost continously

(not shown). We attribute this apparent mismatch to optically thin clouds that are not detected by the ceilometer and may not appear as clouds to a human observer. The continuous 100% cloud cover produced by the AWI-CM3/IFS thus reflects the

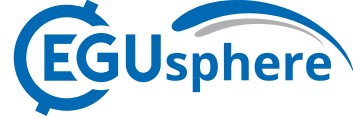

presence of condensate over the MOSAiC site, while the lower and more variable cloud cover in AWI-CM1/ECHAM is closer to the cloud cover perceived by an optical instrument.

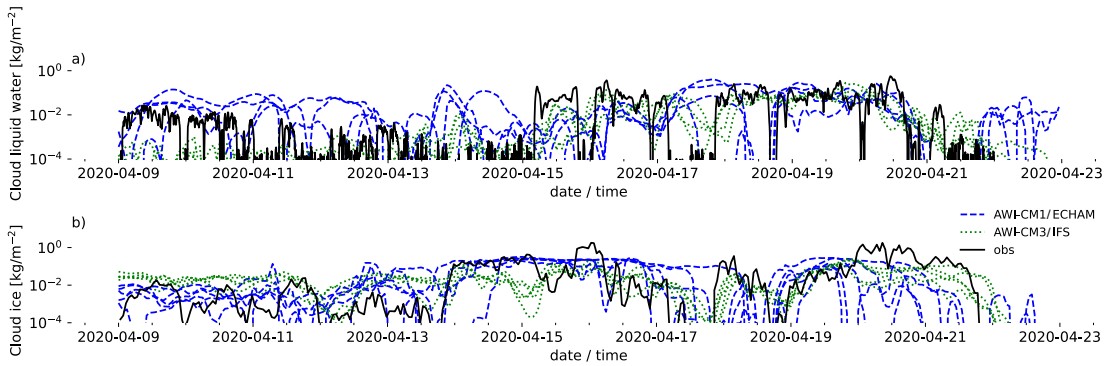

**Figure 5.** Observed and modelled liquid and ice water path for the period 9 to 23 April 2020.

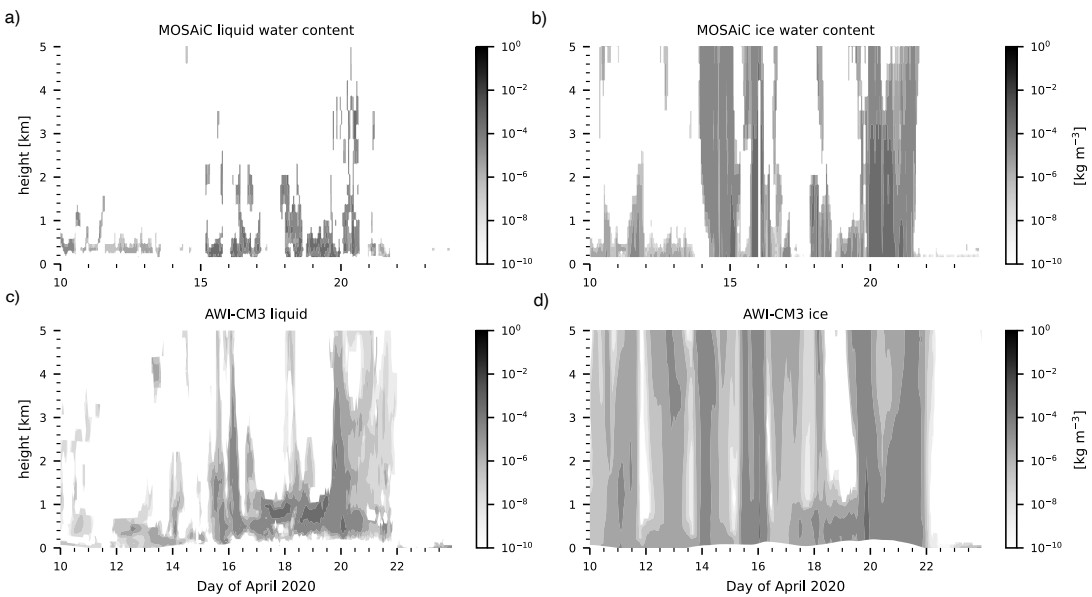

**Figure 6.** Observed and modelled liquid and ice water content (sum of cloud and precipitation) for the period 9 to 23 April 2020.

Throughout much of the cold, dry phase prior to the intrusion, the estimate of the liquid water path in observations is lower than the observational uncertainty of about $10^{-2}$ kgm$^{-2}$ (Fig. 5). AWI-CM1/ECHAM has a substantially higher liquid water path, whereas the liquid water path in AWI-CM3/IFS is about two orders of magnitude smaller than suggested by observations in this period. Both models have more realistic liquid water paths during the intrusion. AWI-CM3/IFS consistently





overestimates cloud ice in the cold phase and AWI-CM1/ECHAM tends to overestimate cloud ice on individual days (eg 12th-14th April).

AWI-CM1's overestimation of surface downwelling shortwave radiation (Fig. 3) in spite of a realistic liquid water path is apparently caused by the partial cloud cover. Note that the liquid water path shown is a grid-box mean value, suggesting that the in-cloud liquid water path in AWI-CM1/ECHAM is even higher. At least in this particular case study, Arctic stratus clouds in AWI-CM1/ECHAM thus mirror a typical pattern of low-latitude stratocumulus cloud biases, which are too few and too bright (Nam et al., 2012).

Both cloud liquid and ice are spread out over substantially deeper layers in AWI-CM3/IFS than in observations (Fig. 6). Other ensemble members (not shown) have very similar profiles of cloud condensate. Tjernström et al. (2021) also reported that clouds in the IFS were too deep in forecasts for a summertime Arctic ocean campaign. Data from AWI-CM1/ECHAM is not shown here as 3-dimensional precipitation is not output in ECHAM and the retrieval does not distinguish between cloud and precipitating condensate.

**3.3 Heat conduction through ice and snow**

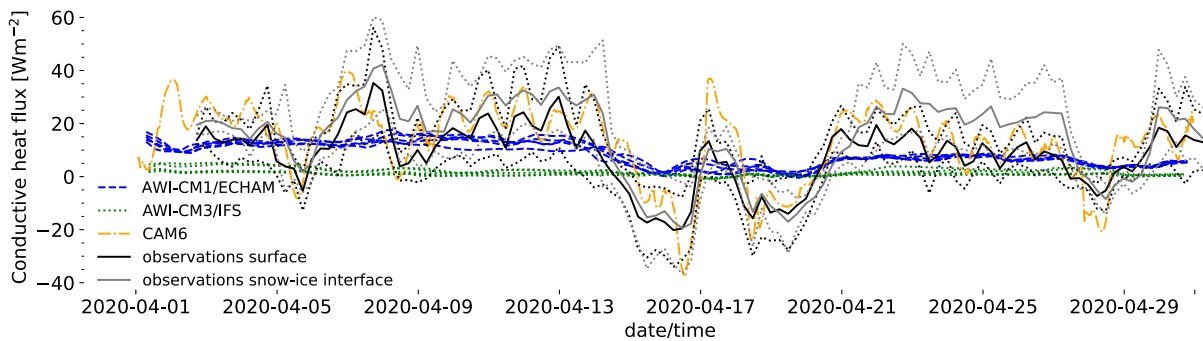

**Figure 7.** Observed and modelled 6-hourly conductive heat fluxes (positive towards the surface, i.e. upwards) during April 2020. Dotted lines show the range of one standard deviation around the mean observational value. Note that measurements are for the uppermost snow layer at the surface and the snow-ice interface, whereas ECHAM data is for conductive heat flux through the ice. For CAM6, the heat flux between the snow surface and the atmosphere was reconstructed from the atmospheric fluxes at the surface.

As the ocean temperature underneath the sea ice is constrained to the freezing point of sea water (-1.9 °C), upward conductive heat flux through the ice makes a noticeable contribution to the seasonally averaged surface energy budget over sea ice in winter. Figure 7 shows observational estimates of heat conduction derived from snow and sea-ice temperatures alongside model output for the conductive heat flux through the ice (AWI-CM1/ECHAM) and the flux passed from the atmosphere to the sea ice (AWI-255 CM3/IFS) for April 2020. These fluxes are not identical. As sea ice has a non-negligible heat capacity, more (less) heat can be conducted upwards at the snow-ice interface than is simultaneously conducted through the ice from the ocean at cold (warm)



surface temperatures, for example. Nevertheless, absolute values and variability of both variables should match approximately over longer time scales, on which the heat capacity of snow and ice is small compared to the energy exchange at the surface.

Both observed and modelled fluxes decrease substantially when the surface temperature rises during the moist intrusion. During the cold period at the beginning of April, fluxes in AWI-CM1/ECHAM are on the order of 15 $Wm^{-2}$, somewhat lower than the mean observational estimate but within one standard deviation. In AWI-CM3/IFS, the conductive heat flux towards the surface is substantially smaller, around 5 $Wm^{-2}$. CAM6 fluxes closely match observed fluxes including a realistic representation of the diurnal cycle as discussed above.

When the MOSAiC region is impacted by the moist intrusions in mid-April, the observed conductive heat flux at the surface and snow-ice interface changes sign and becomes negative. There is a convergence of conductive heat flux within the snow-ice system, and the snow pack and sea ice are warmed by the atmosphere. The output from AWI-CM1/ECHAM is not expected to reflect this downward flux, as the conductive heat flux through the ice cannot become negative - the ice would melt before conducting heat to the ocean. In AWI-CM3/IFS, downward fluxes do briefly occur in mid-April, but can hardly be seen in Fig. 7 due to the much smaller magnitude of the flux.

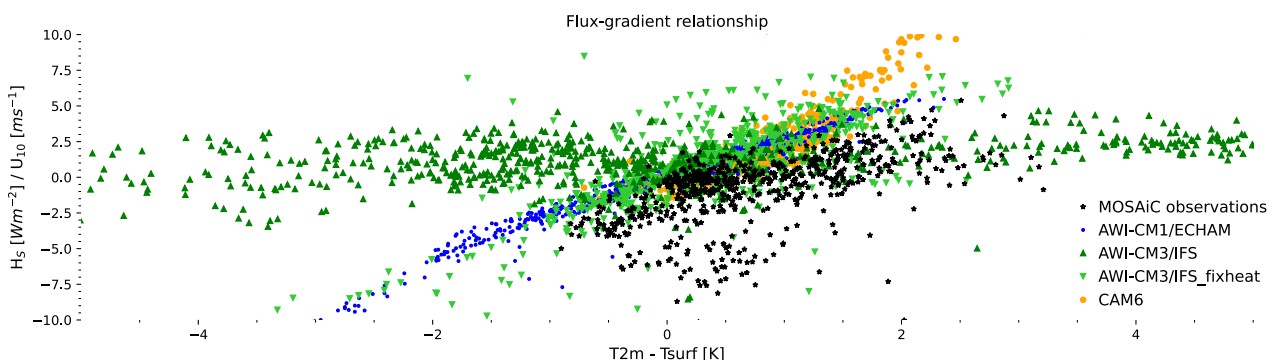

**Figure 8.** Relationship between near-surface temperature gradients and sensible heat fluxes (positive downwards) using hourly averages.

### 3.4 Turbulent heat flux

As neither version of AWI-CM reproduces the observed diurnal cycle with its alternation between a stably stratified and a shallow convective boundary layer, these models have little chance of reproducing the observed sensible heat flux at any moment in time despite the nudging. But models should represent the observed relationship between the near-surface temperature gradient and the heat flux normalized by the wind speed (see Fig. 8, flux is positive downwards,Tjernström et al. (2005)). In this representation, the slope of a regression line corresponds to the transfer coefficient in sensible heat flux parameterizations. Note that we use the diagnostic 2 m temperature and 10 m winds in models to match the observations. Using the lowest level atmospheric temperatures instead would not qualitatively change the conclusions (not shown).





Taking into account some scatter due to measurement errors, the deviation of AWI-CM1/ECHAM from observations corresponds to known deficiencies in its representation of turbulent surface fluxes (Pithan et al., 2015), that are largely representative

for weather prediction and climate models: under strongly stable stratification, i.e. when the 2 m temperature is substantially larger than the surface temperature, sensible heat fluxes towards the surface are overestimated in models. This corresponds to the long-standing issue of models producing too much turbulence in strongly stable boundary layers (Holtslag et al., 2013). Under unstable stratification (negative gradients in Fig. 8), ECHAM produces unrealistically large temperature gradients. This is due to the purely local diffusion scheme in ECHAM that cannot directly represent the mixing by large eddies throughout the

entire boundary layer. Combined eddy-diffusivity-mass-flux (EDMF) schemes represent this more realistically.

In contrast, AWI-CM3/IFS (green triangles in Fig. 8) produces much larger temperature gradients than observed, hardly shows a correlation between the temperature gradient and sensible heat flux, and frequently produces downward turbulent fluxes despite a negative gradient (i.e. values in the upper left quadrant of Fig. 8). We attribute this to an inconsistent treatment of surface coupling: the IFS uses separate skin and surface temperatures, whereas surface temperatures in the coupled

model are updated on the FESOM side using a scheme modelled after ECHAM6, which does not distinguish skin and surface temperatures. Effectively enforcing $T_{skin} = T_{surface}$ in the IFS by setting the skin layer conductivity to $10^{10}$ Wm$^{-2}$K$^{-1}$ (as discussed in Hartung et al. (2022)) largely fixes this issue (downward pointing triangles in Fig. 8). The spread is still larger than in AWI-CM1/ECHAM, which we attribute to less frequent updates of the surface temperature (every two hours in AWI-CM3/IFS vs. each timestep in AWI-CM1/ECHAM). This substantial improvement of the turbulent surface fluxes only has

a small impact on the overall temperature evolution (not shown), suggesting compensating errors in other fluxes. The large modelled thermal inertia of the snowpack may also contribute to this small impact of changes in the surface flux computation on temperatures.

CAM6 (orange circles in Fig. 8) correctly represents the cutoff of the temperature gradient in unstable situations, i.e. when the surface is warmer than the atmosphere, but overestimates downward heat fluxes under stable stratification even more than

AWI-CM1/ECHAM and AWI-CM3/IFS.

### 3.5 Temperature profiles before and during the intrusion

Temperature inversions, i.e. temperature increasing with height, frequently occur in the lower Arctic troposphere in the cold season (Serreze et al., 1992). They play an important role for the radiative effect of clouds (Sedlar et al., 2012) and for the lapse-rate feedback, which is an important contributor to Arctic amplification (Manabe and Wetherald, 1975; Pithan and Mauritsen,

2014). CMIP3 and CMIP5 climate models had substantial biases in the typical Arctic wintertime inversion strength (Medeiros et al., 2011; Pithan et al., 2014). In the following, we analyse how the AWI-CM models represent the atmospheric temperature profile on two days chosen to represent the cold, dry air phase at the beginning of April and the moist intrusion.

The sounding from 11 April shows a cold air mass with elevated temperature inversions (Fig 9). The inversion layer is interrupted by a thin cloud layer, with strong and deep inversions below and above the cloud. Cloud-driven mixed layers that

do not reach down to the surface called decoupled clouds and clouds capped by or extending into a temperature inversion have



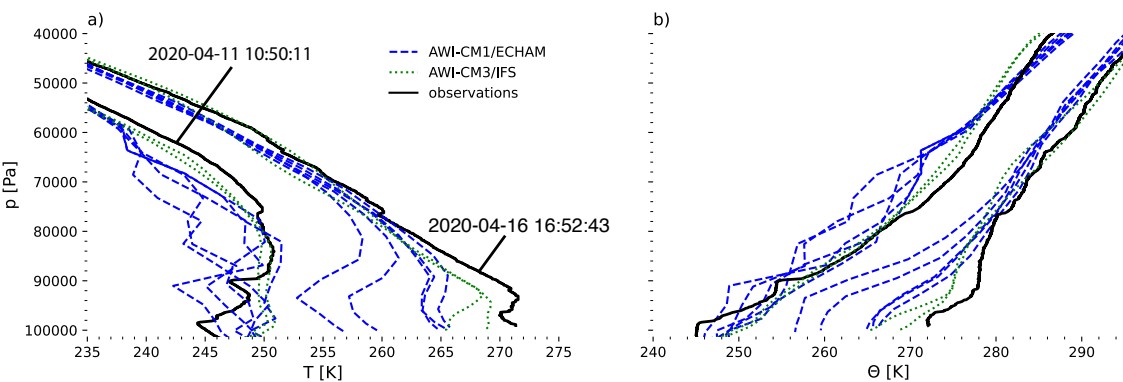

**Figure 9.** Modelled and observed a) temperature and b) potential temperature profiles at the MOSAiC site before (2020-04-11T10:50:11) and during (2020-04-16T16:52:43) the moist intrusion. High-frequency model-level output was only saved for two of the AWI-CM3/IFS ensemble members, but these cover the full spread of the five-member ensemble (not shown).

been described before (Sedlar et al., 2012; Shupe et al., 2013). Here, the cloud-driven mixed layer is particularly thin, which points to cloud-driven turbulence being weak compared to the atmospheric stratification.

Both model versions also show a cold air mass and at least one temperature inversion at the ground or aloft, with AWI-CM1 substantially underestimating low-level atmospheric temperatures. AWI-CM3/IFS does not reproduce the elevated mixed layer
or temperature inversion at all, but has a rather steady increase in potential temperature with height. More and less stable layers are apparent in AWI-CM1/ECHAM, but the model does not form an obvious cloud-driven mixed layer with constant potential temperature as seen in observations. These temperature profiles are consistent with the representation of clouds and cloud condensate discussed above: A high-emissivity liquid layer as present in AWI-CM1/ECHAM may be required to generate the elevated mixed layer and inversion layer, which may not be as pronounced in AWI-CM1/ECHAM as the model only has partial
cloud cover in this period. The deeper ice cloud in AWI-CM3/IFS produces realistic shortwave radiation at the surface, but lacks a single high-emissivity layer in the atmosphere that could sustain a mixed layer and elevated temperature inversion. The shallow mixed layer observed close to the ground is probably caused by solar heating of the surface, and is not captured by the AWI models due to their excessive latency in surface temperature discussed above.

During the moist intrusion on 16 April, the observed temperature profile is largely adiabatic in the lower troposphere with a
shallow elevated temperature inversion of a few Kelvin. While AWI-CM3 matches the profile rather well, AWI-CM1 substantially underestimates atmospheric temperature near the surface and slightly underestimates it above the boundary layer. The near-surface cold bias on 16 April is at least partly related to the brief cooling between the main warm pulses being somewhat too early in AWI-CM1/ECHAM (see Fig. 2). A cold bias in the free troposphere is also visible in the profile from 11 April and in temperature profiles from Ny-Alesund, i.e. close to the sea-ice edge during cold-air advection from the central Arctic, but
not during warm air advection (not shown). This suggests that AWI-CM1/ECHAM computes excessive atmospheric cooling rates in the free troposphere during the transformation of initially warm and moist air masses over sea ice.



## 4   Conclusions

Our case study uses high-frequency in-situ data from the MOSAiC expedition and nudging of the large-scale circulation of climate models towards ERA5 in the free troposphere. The study shows that nudging a coupled model is an effective way to
directly evaluate climate model physics with observations, even for individual events.

Both the atmosphere-ice-ocean models AWI-CM1/ECHAM and AWI-CM3/IFS and the atmosphere-only model CAM6 reproduce the key features of the observed cold phase at the beginning of April 2020, the warm phase dominated by moist intrusions mid-April and subsequent cooling thereafter. Under optically thin clouds during the cold phase, a clear diurnal cycle is observed but not captured by the AWI-CM models, which we attribute to the simplistic treatment of the snow pack as a
single layer with uniform temperature in both models. CAM6 uses three layers to represent snow on sea ice (Danabasoglu et al., 2020) and captures the diurnal cycle much better than the AWI-CM models.

AWI-CM1/ECHAM correctly models the occurrence of cloud liquid water during this period, but overestimates the liquid water path and underestimates cloud cover, which is nearly ubiquitous in observations. As a result, the model overestimates downwelling shortwave radiation at the surface. AWI-CM3/IFS has about two orders of magnitudes less cloud liquid wa-
ter, but a fully overcast sky and surface downwelling shortwave radiation close to observed, likely due to compensation by overestimated cloud ice. CAM6 closely matches observed surface shortwave radiation. Cloud phase in the present-day climate controls the potential for future cloud brightening as ice clouds transition to optically thicker liquid clouds, an important climate feedback with substantial impacts on Earth's climate sensitivity (Ceppi et al., 2017; Zelinka et al., 2020).

In the original setup, we detected an unphysical relationship between the near-surface temperature gradient and turbulent
surface fluxes in AWI-CM3/IFS. We resolved this issue by making the treatment of surface/skin temperature more consistent with the surface temperature update routine. Within our case study, all models overestimate turbulent heat fluxes under stable stratification, and this overestimation is stronger in CAM6 than in AWI-CM1/ECHAM and AWI-CM3/IFS.

Our evaluation underscores the need to further improve the model representation of mixed-phase clouds in cold environments and of stable boundary layers. It suggests that going beyond a 1-layer model for the representation of snow thermodynamics
over sea ice would be beneficial for ice-atmosphere coupling in climate models.

Observing system simulators (Zhang et al., 2018) would facilitate even closer comparisons to cloud radar and lidar data. For studies in the polar regions involving spatially narrow features such as moist intrusions, we recommend the use of strong nudging with relaxation timescales on the order of 1h to limit ensemble spread and stay as close to the observed large-scale flow as possible.

We conclude that nudging provides a strong tool to leverage observations, especially from intense, time-limited campaigns, for the evaluation and improvement of coupled climate models. A nudging intercomparison involving more coupled models and using the full MOSAiC dataset, data from the COMBLE campaign (Geerts et al., 2022), YOPPsiteMIP (Uttal et al., 2019) and recent Southern ocean campaigns (McFarquhar et al., 2021) would be an asset for evaluating and improving the representation of crucial processes in climate models.





*Code and data availability.* MOSAiC observations are available from multiple sources. The Met City flux tower, meteorological data, and snow depth are available at the Arctic Data Center (https://doi.org/10.18739/A2VM42Z5F, 2021, Cox et al. (2021)). The Met City radiation measurements (https://doi.org/10.5439/1608608, Riihimaki (2019)), ceilometer data (https://doi.org/10.5439/1181954, Morris et al. (2021), and ShupeTurner cloud microphysics product (https://doi.org/10.5439/1871015, Shupe (2022)) are available from the DOE Atmospheric Radiation Measurement archive. The radiosonde data are available from the PANGAEA archive (https://doi.org/10.1594/PANGAEA.928656, Maturilli et al. (2021)). Snow temperature measurements from SIMBAs are available from the PANGAEA archive (see Table 1). MOSAiC buoy data is available at https://data.meereisportal.de/data/buoys/processed/MOSAiC/mosaic_buoy_data.zip.

| https://doi.pangaea.de/10.1594/PANGAEA.940393 | 940231 | 940593 | 940617 | 940634 | 940659 |
|---|---|---|---|---|---|
| | 940668 | 940680 | 940692 | 940749 | 940702 |

**Table 1.** DOIs for SIMBA snow temperature datasets. The last six digits change for each SIMBA.

CESM/CAM6 model code is available at https://github.com/ESCOMP/CESM. The ocean model FESOM2 source code is available on Zenodo at 10.5281/zenodo.6335383 and at https://github.com/FESOM/fesom2/releases/tag/AWI-CM3_v3.0. OpenIFS is not publicly available but rather subject to licecing by ECMWF. However licences are readily given free of charge to any academic or research institute. All modifications required to enable AWI-CM3 simulations with OpenIFS CY43R3V1 as provided by ECMWF can be obtained on Zenodo at: 10.5281/zenodo.6335498. The OASIS coupler is available upon registration at: https://oasis.cerfacs.fr/en/downloads/. The XIOS source code is available on Zenodo (10.5281/zenodo.4905653, Meurdesoif, 2017) and on the official repository (http://forge.ipsl.jussieu.fr/ioserver, last access: 4 March 2022). The runoff mapper scheme is available on Zenodo at 10.5281/zenodo.6335474. The compile and runtime engine esm-tools is available on Zenodo at: 10.5281/zenodo.6335309.

*Author contributions.* F.P. conceived the study, analysed the data, produced the figures and wrote the manuscript with input from all other authors. A.S.-B. and M.A. performed nudged model runs with AWI-CM1 and AWI-CM3. J.S. implemented and tested the corrected surface coupling in AWI-CM3. A.S. computed the conductive heat flux estimate from SIMBA temperatures. M.D.S. processed observational data. S.D. compiled and quality-checked the radiosonde data. All authors contributed to and commented on the manuscript.

*Competing interests.* The authors declare no competing interests.

*Acknowledgements.* Data used in this manuscript were produced as part of the international Multidisciplinary drifting Observatory for the Study of Arctic Climate (MOSAiC) with tag MOSAiC20192020. We thank all persons involved in the expedition of the Research Vessel Polarstern during MOSAiC in 2019-2020 (AWI_PS122_00). Radiation and ceilometer data were obtained from the Atmospheric Radiation Measurement (ARM) User Facility, a U.S. Department of Energy (DOE) Office of Science User Facility Managed by the Biological and Environmental Research Program. Radiosonde data were obtained through a partnership between the leading Alfred Wegener Institute (AWI), the atmospheric radiation measurement (ARM) user facility, a US Department of Energy facility managed by the Biological and



Environmental Research Program, and the German Weather Service (DWD). F.P. acknowledges funding from the European Union's Hori-zon 2020 research and innovation programme under grant agreement No 101003826 via project CRiceS (Climate Relevant interactions and feedbacks: the key role of sea ice and Snow in the polar and global climate system). M.D.S. was supported by the DOE Atmospheric System Research Program (DE-SC0021341, DE-SC0019251), U.S. National Science Foundation (OPP-1724551), and NOAA Physical Sciences
395     Laboratory. M.A. acknowledges funding by the Federal Ministry of Education and Research of Germany (BMBF) in the framework of SSIP (grant01LN1701A). J.S. was supported by project L4 of the Collaborative Research Centre TRR 181 "Energy Transfers in Atmosphere and Ocean" funded by the Deutsche Forschungsgemeinschaft (DFG, German Research Foundation) under Project 274762653. A.S.-B. acknowl-edges funding by the Federal Ministry of Education and Research (BMBF) and the Helmholtz Research Field Earth & Environment for the Innovation Pool Project SCENIC. Colin Zarzycki is acknowledged for providing the CAM simulation that is together with GS perfomed
400     as part of work supported by a Climate Process Team (CPT) under Grant AGS-1916689 from the National Science Foundation and Grant NA19OAR4310363 from the National Oceanic and Atmospheric Administration. CAM simulations were completed using high-performance computing support from Cheyenne (doi:10.5065/D6RX99HX) provided by NCAR's Computational and Information Systems Laboratory, sponsored by the National Science Foundation. Autonomous sea ice measurements (lightchain buoy 2020R11) from 2020-4-1 to 2020-4-29 were obtained from https://www.meereisportal.de (grant: REKLIM-2013-04).



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
