# Peer review of "Nudging allows direct evaluation of coupled climate models with in-situ observations: A case study from the MOSAiC expedition"

_EGUsphere, 2022_

## Author Response (AR1)

We would like to thank both reviewers for their positive evaluation of our work, thorough reading of the manuscript and the detailed and helpful suggestions for further improvements.

*This manuscript describes evaluation of two coupled climate models and one general circulation model against detailed in-situ and remote sensing observations in the Arctic. The manuscript is generally well written, and the topic is appropriate and interesting for EGUsphere. The methods for nudging the coupled system are interesting and will be useful for others. I think the manuscript should be publishable subject to minor revisions, detailed in specific comments below.*

*Generally, I think some parts of the text could be more specific in the treatment of the different models. Also, some of the figure presentations could be improved or clarified.*

*Page 1, L12: what does CAM6 show?*

Unfortunately, cloud cover was not saved in the CAM runs analyzed here. We added a note on this to Figure 1 in the revised paper.

*Page 2, L42: Add Gettelman et al 2020, since they also used CAM6 in the S. Ocean.*

Thanks for the reference! We have added it to the following paragraph on nudging applications.

*Page 3, L88: If CAM6 is nudged, you also need to cite Gettelman et al 2020 who nudged CAM6 and looked at the S. Ocean.*

We have added the reference to the introduction (see above), but keep the focus on the model documentation papers in this section.

*Gettelman, A., C. G. Bardeen, C. S. McCluskey, E. Järvinen, J. Stith, C. Bretherton, G. McFarquhar, C. Twohy, J. D'Alessandro, and W. Wu. "Simulating Observations of Southern Ocean Clouds and Implications for Climate." Journal of Geophysical Research: Atmospheres 125, no. 21 (2020): e2020JD032619. https://doi.org/10.1029/2020JD032619.*

*Page 5, L107: CAM6 cloud fraction?*

Unfortunately, cloud cover was not saved in the CAM runs analyzed here. We added a note on this to Figure 1 in the revised paper.

*Page 5, L126: This is confusing. Did you use 1 hour or 24 hour for CM3? Also, the all wavenumber case for CM3: was that 1 hour or 24hour?*

We have split the paragraph into one describing the setup used in this study and one on other settings that we tested and discarded to avoid any confusion in the revised manuscript. We also specify that the 'strong nudging' test with AWI-CM3 used a 1-hour relaxation timescale.

*Page 6, L137: what is the nudging timescale in CAM6?*

The nudging timescale is 1h, added to the revised manuscript. Thanks for pointing out the omission!

*Page 9, Figure 3: Describe model lines and gray line in 3d in caption.*

done

*Page 10, L216: CAM6*

*corrected, thanks*

*Page 10, L218: is the CAM6 albedo shown in Fig 3e? I cannot see it.*

It was indeed missing and has been added in the revised manuscript. Thanks!

*Page 10, L230: Can you put then an estimate of observed cloud cover on Figure 3f? Or is it 100% since the radar has cloud condensate all the time?*

According to the radar, cloud cover should be 100 %, according to the ceilometer, it should be considerably less. We discuss this in the text (section 3.2) but decided that adding these observational estimates to the Figure would not be helpful.

*Page 10, L231: would the models detect such a cloud?*

In principle, models should be able to produce a cloud with a low LWP – in practice, that cloud is likely to be thin not just optically but also in terms of the vertical extent, such that vertical resolution might be a limiting factor.

*Page 11, L233: is there condensate in CM1?*

Yes, condensate is discussed in the following paragraph.

*Page 11, Fig 6: why no CM1 and CAM6?*

3d precipitation fields were only output by AWI-CM3. We have added a note to the caption.

*Page 12, L249: What about CAM6? If it gets the radiation right, does it get the condensate right too? Please be consistent in the treatments here (e.g. CAM6 is on Figure 7 but not Figures 5 & 6, why?)*

We have added CAM6 to Figure 5 (vertically integrated condensate was not available, but could be derived) and the corresponding discussion. Interestingly, CAM6 appears to get SW down right without cloud liquid and without a clear compensating overestimation of cloud ice.

*Page 12, Figure 7: its hard to distinguish the gray dotted lines from the model dotted lines. Perhaps use a shaded region for the standard deviation.*

Good point, we use a shaded region in the revised paper.

*Page 12, L255: why are they not equal? Change of temperature of the ice itself?*

Indeed, this is explained in the following sentence. We swapped the . for – to improve the flow of the text.

*Page 13, L275: shouldn't the relationship be tighter then in figure 8 if it represents a coefficient in the models?*

Some spread is to be expected, not least because the plotted data is averaged over several model time steps. We have edited the sentence to read 'the transfer coefficient computed in sensible heat flux parameterizations' to avoid the possible misinterpretation of the transfer coefficient as a fixed model parameter.

*Page 14, L286: Dark green triangles.*

Thanks, corrected.

*Page 14, L307: Why not include CAM6?*

We do not include the CAM6 temperature profiles here because T above 690 hPa is nudged in the CAM6 runs, so that model biases in free-tropospheric temperature cannot develop freely.

*Page 15, L311: These models are nudged all the way to the surface. How much does that matter? What does CAM6 (not nudged below 690hPa) do? Does the spectral nudging matter?*

Vorticity and divergence in the AWI models are only nudged above 700 hPa, and the thermodynamic variables aren't nudged at all. CAM6 has a pretty realistic T profile (but without the cloud-driven mixed layer). See above on why CAM6 is not included in the plot.

*Page 15, L318: can you refer to Fig 6 and put a line for the time of the sounding to guide the reader here?*

Good point, we have added such lines to Figures 5 and 6

*Page 16, L358: what does CAM6 do for liquid water? Gettelman et al 2020 showed overestimations of supercooled liquid in the S. Ocean.*

L. 346: We have expanded the statement as follows: "CAM6 closely matches observed surface shortwave radiation despite lacking cloud liquid water. In contrast to AWI-CM1/IFS, CAM6 does not have a clear overestimation of cloud ice, so the reasons for the close match of the shortwave radiation is unclear."

*Page 16, L361: Can you say anything about the type of nudging? E.g. spectral v. Full nudging, and vorticity/divergence v. Winds & Temps. Does it matter? What is better? How can you tell?*

We have added the sentence 'Nudging intensity needs to strike a balance between constraining the relevant weather phenomena and leaving the model sufficient freedom to

respond in a physically plausible way.' The range of setups and models tested here may be too small to be more specific.

- **Citation**: *https://doi.org/10.5194/egusphere-2022-706-RC1*
- **RC2**: *'Comment on egusphere-2022-706', Anonymous Referee #2, 20 Jan 2023 reply*

*Peer review for manuscript: Nudging allows direct evaluation of coupled climate models with in-situ observations: A case study from the MOSAiC expedition*

*Climate models are important tools for understanding complex interactions of physics and dynamics in the atmosphere. Descriptions of dynamical and especially physical processes are not complete in the models and simplifications made in parametrizations of physical processes causes uncertainty in model results. Therefore, it is very important to evaluate accuracy of models.*

*Typically accuracy of climate models is estimated by comparing model results against long time series of observations (or data sets which are strongly constrained by observations as reanalyses). As the state of large circulation, which causes a large part of variability of atmospheric conditions in middle and high latitudes, in models are not connected to state of large circulation in the real atmosphere, long time series is needed for comparisons, so that they are able to present a sufficient part of climatological variability which is caused by variations in large scale circulation. The requirement of long time series rules out a lot of observational data which otherwise could be used for evaluation of climate models. Especially in high latitudes, where the permanent observational network is sparse and a lot of observations have been collected from relatively short measurement campaigns. This strongly limits the evaluation of accuracy on climate models in the high latitudes where the presentation of atmospheric conditions in climate models is often worse than in lower latitudes. Therefore, it is highly important to evaluate the capability of climate models to simulate polar climate, which further allows development of models, so that they can better simulate atmospheric conditions also in the high latitudes. The manuscript presents a method how to overcome the requirement of long time series using nudged model simulations. When nudged simulations are utilised, large scale circulation in the model is strongly constrained by the observed large scale circulation. This kind of methodology allows direct comparison of model results and observations. As the large circulation in nudged simulation is constrained by the observed large scale circulation, model physics are responsible for uncertainty of models. This method does not allow estimate biases which are associated with presentation of large scale circulation in the models, but often a large part of the uncertainty is associated with parameterized physical processes. However, the possibility to use short time series of observation provides large advantages for model evaluations and further model development.*

*The novelty of the manuscript is in its methodology. The set of models that is evaluated in the manuscript is not comprehensive by any means, but I think that is not the scope of the manuscript. However, even this set of models shows interesting differences in their capability to simulate atmospheric conditions and clearly demonstrate the usefulness of nudged simulation for model evaluation and also shows some deficiencies in models especially associated with treatment of snow surface. Therefore, the scientific value of this manuscript supports the publishing of the manuscript in Geoscientific Model Development.*

**General comments:**

*1) Nudging is probably familiar for many readers, but I still suggest adding a short general description of nudging in the beginning of the nudging paragraph in the method section. Does nudging cause artificial effects on time series when model state is nudged towards real atmospheric state?*

We briefly introduce the concept of nudging in the introduction, and have added the following statement: "Over longer timescales (years to decades), nudged coupled models tend to develop climatological biases that are not the same as in their free-running equivalents. This issue is not a first-order problem for the relatively short runs analysed here, and not further explored in this paper."

*2) As model biases are related to weather conditions, I would start the result section with short description of weather conditions where, in addition to temperature changes, you could shortly describe e.g. cloud conditions, cloud cover and cloud liquid and ice water content, stratification, longwave and shortwave radiative fluxes, turbulent heat fluxes and how they are related.*

We have added such a paragraph. Thanks for suggesting this!

*3) The differences in surface temperature and skin temperature between the models are well explained in the method section, but it is still sometimes tricky to follow which is meant by surface/skin temperature in some parts in the results section. Therefore, I would suggest paying attention to clarity when surface/skin temperature is discussed in the result section and add remainder about their meanings when it is not very clear which temperature is in question.*

Thanks for pointing out this lack of clarity, we have revised the results section accordingly.

*4) Longwave radiation often has a remarkable effect on surface and near surface temperatures. However it has not received a lot of attention in the manuscript. If you have observational data of longwave radiative fluxes, I would suggest adding comparison of longwave radiative surface fluxes between models and observations, and how differences in cloud cover and cloud water (liquid and ice) content affect longwave radiative fluxes as well as how differnces in longwave radiative fluxes affect surface temperatures.*

Downwelling longwave radiation is so dependent on small changes in cloud properties that it fluctuates at short time scales and is not sufficiently constrained by the nudging for a meaningful 1:1 comparison. A statistical evaluation would of course be feasible, but we here want to focus on rather simple diagnostics that show the added value of nudging.

***Specific comments:***

*Lines 100 – 101, Are the different values for rhcrit and rhsat used for the whole column below 700hPa or only in the inversion layer if a temperature inversion exists below 700 hPa?*

These changes apply below the inversion. We make this clear in the revised manuscript.

*Lines 141 – 144, How representative observations are for the whole grid cells used in comparisons? Were the conditions in the area of grid cell homogeneous enough (e.g.*

*occurrence open water causes inhomogeneity) that point measurement could represent average condition in the grid cell?*

In the revised manuscript, we state "We do not explicitly address the heterogeneity that may exist within the scale of a model grid box in the present study, but our use of hourly averages means that any non-stationary inhomogeneities on scales ~ 10 km along-wind would be advected across the measurement site and averaged out." Sea-ice cover around MOSAiC was still closed in April and is virtually closed in the models used here.

*Lines 188 – 189, How coarser resolution affects the delay?*

Revised: "We attribute this to the coarser horizontal resolution in the region compared to AWI-CM1/ECHAM (see Fig. \ref{fig_map}), as a newly-arriving arriving air mass may have to travel further to reach the closest grid-point than to the actual MOSAiC location.

*Line 195 Do you compare surface temperatures from models against observed skin temperature? The next paragraph maybe gives an answer, but maybe it is good to clarify also here.*

Clarified in the revised manuscript, thanks!

*Line 270, Which kind of weather conditions are associated with unstable stratification in AWI-CM models?*

We did not investigate when unstable stratification occurred in the model.

*Lines 286 – 297, Have you calculated the relationship between sensible heat flux and difference between skin temperature and 2m temperature in AWI-CM3/IFS. How would it look? Maybe the relationship between sensible heat flux and difference between skin temperature and 2m temperature looks better than relationship between sensible heat flux and difference between surface temperature and 2m temperature because in the method section it has been mentioned that AWI-CM3/IFS uses skin temperature for turbulent heat fluxes.*

The skin temperature computed in the IFS was overwritten by the coupling routine and therefore couldn't be output.

*Lines 330 – 331, Could stronger cooling in AWI-CM1/ECHAM be associated with clouds?*

That was indeed our initial hypothesis supported by some snapshots of cloud condensate profiles, but these results weren't fully consistent over time. To be continued…

*Lines 344 – 346, Has the overestimation of cloud ice content so large an impact on radiation that it can compensate the underestimation of cloud liquid water content?*

We haven't formally tested that, but do see this as the most plausible explanation.

*Overall, the manuscript is well written conclusion based on evidence of results. Methods are appropriately described allowing readers to understand how study is done. Therefore, I*

*recommend publishing the manuscript in Geoscientific Model Development after minor revision.*

---

## Author Response (AR2)

*Thank you for addressing the specific comments from both reviewers. I am happy to accept this manuscript for publication in GMD. I would only ask that you provide a short sentence in the manuscript summarizing your response to Reviewer 2's point 4, explaining why LW nudging was not done. This comment might be useful for others trying to reproduce or build on this work in the future.*

We have added the following paragraph to the results section:

Downwelling and surface net longwave radiation play an important role in determining the surface energy balance and boundary-layer state in Arctic winter (Stramler et al., 2011; Pithan et al., 2014). During the period studied here, downwelling longwave radiation fluctuates on short time scales in both models and observations, which we attribute to subtle changes in cloud properties that are not constrained by the large-scale nudging (not shown). Longwave radiation could still be evaluated using process-based metrics (Pithan et al., 2014), but not with a one-to-one comparison of hourly averages as shown for the shortwave fluxes.